# Characteristics of Cellulose Nanofibrils from Transgenic Trees with Reduced Expression of Cellulose Synthase Interacting 1

**DOI:** 10.3390/nano12193448

**Published:** 2022-10-02

**Authors:** Simon Jonasson, Anne Bünder, Linn Berglund, Totte Niittylä, Kristiina Oksman

**Affiliations:** 1Division of Materials Science, Luleå University of Technology, 97187 Luleå, Sweden; 2Umeå Plant Science Centre, Department of Forest Genetics and Plant Physiology, Swedish University of Agricultural Sciences, 90183 Umeå, Sweden; 3Mechanical & Industrial Engineering, University of Toronto, Toronto, ON M5S 3G8, Canada; 4Wallenberg Wood Science Centre (WWSC), Luleå University of Technology, 97187 Luleå, Sweden

**Keywords:** transgenic wood, cellulose nanofibrils, fibrillation, network properties

## Abstract

Cellulose nanofibrils can be derived from the native load-bearing cellulose microfibrils in wood. These microfibrils are synthesized by a cellulose synthase enzyme complex that resides in the plasma membrane of developing wood cells. It was previously shown that transgenic hybrid aspen trees with reduced expression of CSI1 have different wood mechanics and cellulose microfibril properties. We hypothesized that these changes in the native cellulose may affect the quality of the corresponding nanofibrils. To test this hypothesis, wood from wild-type and transgenic trees with reduced expression of CSI1 was subjected to oxidative nanofibril isolation. The transgenic wood-extracted nanofibrils exhibited a significantly lower suspension viscosity and estimated surface area than the wild-type nanofibrils. Furthermore, the nanofibril networks manufactured from the transgenics exhibited high stiffness, as well as reduced water uptake, tensile strength, strain-to-break, and degree of polymerization. Presumably, the difference in wood properties caused by the decreased expression of CSI1 resulted in nanofibrils with distinctive qualities. The observed changes in the physicochemical properties suggest that the differences were caused by changes in the apparent nanofibril aspect ratio and surface accessibility. This study demonstrates the possibility of influencing wood-derived nanofibril quality through the genetic engineering of trees.

## 1. Introduction

Wood is a complex biological structure that provides trees with the mechanical support needed to ensure upright growth while simultaneously facilitating the transport of nutrients and water [1]. Mechanical reinforcement in wood is performed by cellulose microfibrils (CMFs), which are hierarchical structures of β(1→4)-linked D-glucose units that form glucan chains of nanoscale semi-crystalline fibrils in a bottom-up fashion [2]. CMFs are positioned throughout the wood cell walls, where they are embedded in a hydrated matrix composed primarily of hemicellulose and lignin [3]. The resulting natural nanocomposite structure gives rise to the distinct properties of wood [4].

The prospect of extracting CMFs from wood as a value-added product has gained attention over the past two decades. CMFs that have been subjected to isolation and purification processes are referred to as cellulose nanofibrils (CNFs) and inherit many of their distinct properties, including high tensile strength [5], stiffness [6], and aspect ratio [7] and a low coefficient of thermal expansion [8].

The development of various processing techniques has further enabled the successful isolation of CNFs, which closely resemble the most basic crystalline structures of CMFs [9,10,11]. Through careful processing, it is also possible to retain a high degree of polymerization (DP) of cellulose [10], which is manifested as longer CNFs [12,13]. These process developments have led to the prospect that the characteristics of CMFs potentially influence isolated CNFs. Thus, understanding how to alter the CMFs in the original feedstock opens the possibility of controlling the final CNF characteristics.

Understanding how CMFs are synthesized in plants is one of the main challenges in plant biology [14]. Research on CMF synthesis machinery has revealed the involvement of multiple proteins in this process [15]. One of the cellulose biosynthesis-associated proteins that plays a crucial role in the alignment of the nascent CMFs into the cell wall is the cellulose synthase interactive 1 (CSI1) protein [16,17]. The plasma membrane-localized cellulose synthase complex (CSC) moves into the plasma membrane during cellulose biosynthesis [18]. CSI1 guides the CSC along the cortical microtubules (cMTs) to align CMFs during primary cell wall biosynthesis [17,19] and during the initial phase of secondary cell wall formation [20]. In a study on transgenic trees with reduced expression of CSI1, it was observed that both the stiffness and strength of the wood decreased, as did the cellulose DP [21]. There were no apparent anatomical or compositional changes in the wood of the transgenic lines, which led to the hypothesis that a reduction in CSI1 may affect the mechanical properties of the wood by reducing cellulose DP, thus altering the CMF characteristics [21]. Thus, it is of interest to isolate CNFs from transgenic trees with reduced CSI1 levels and assess the influence of genetic modification on CNF properties.

To compare CNFs isolated from transgenic wood (T) with wild-type (WT) and mild (pH = 6.8) direct oxidation using the catalytic system, 2,2,6,6-tetramethylpiperidin-1-yl) oxyl (TEMPO) was employed together with the mechanical separation process. This process allows for the direct oxidation of lignin and cellulose in one experimental step while preserving the cellulose DP [10,22,23,24,25]. Two transgenic lines with reduced expression of CSI1 were studied together with the control tree (WT) with normal CSI1 expression. The isolated CNFs were characterized in the dispersion state using viscosity, conductimetric titration, atomic force microscopy, and yield- and surface-area estimations. The networks were then manufactured from the dispersions and tested for DP, water uptake, and mechanical behavior. The characteristics of CNFs and their networks are discussed in the context of the initial wood properties.

## 2. Materials and Methods

### 2.1. Materials

Wood from greenhouse-grown hybrid aspen trees (*Populus tremula x tremuloides*) was used as feedstock for CNF isolation. Two transgenic lines (T1 and T2) with significantly reduced expression of *CSI1* and wild-type (WT) trees as controls were used in this study. The details of the genetic modification and characterization of these trees and their wood chemistry and properties were described by Bünder et al. [21]. To obtain wood for CNF isolation and analysis, in vitro micro-propagated wild-type and transgenic hybrid aspen trees were transferred to soil and grown in a greenhouse under an 18 h light/6 h dark photoperiod at a temperature of 22 °C/15 °C (light/dark) and 50–70% humidity. The trees were harvested after eight weeks of growth at approximately 160 cm in height. Stem wood from five to six biological replicate trees per genotype was pooled to obtain sufficient material for CNF isolation. The physical and chemical characteristics of the T1, T2 (corresponding to CSI1RNAi-1 and CSI1RNAi-3, respectively), and WT, were determined by Bünder et al. [21], and there were no significant differences in the lignin or carbohydrate contents, whereas the mechanical properties of the WT were better than those of the transgenic wood samples.

Sodium chlorite (77.5–82.5%, high purity), standard hydrochloric acid (0.5 N), standard sodium hydroxide solution (0.1 N), and sodium hydroxide beads (>97%, ACS) were purchased from VWR, Solna, Sweden. TEMPO (99%), sodium hypochlorite (NaClO, 6–14% active chlorine), Congo red, and 1 M copper (II) ethylenediamine solution were purchased from Sigma-Aldrich, Stockholm, Sweden AB. All chemicals were used as received.

### 2.2. Methods

#### 2.2.1. Wood Oxidation and Fibrillation

The wood samples were ground and sieved through a 300–500 µm mesh to obtain powders with uniform particle sizes. The wood powder (2 g per sample) was dried in an oven overnight and then soaked in distilled water for 24 h prior to starting oxidation. Sodium chlorite (5.0 g/g wood) was added together with TEMPO to the vessel (250 mL) containing the soaked wood powder and a phosphate buffer (pH = 6.8). The vessels were submerged in a shaking water bath (Cole-Parmer Stuart, Staffordshire, UK) at 60 °C for 1 h to dissolve the reagents. The reaction was started by adding sodium hypochlorite (2 mL/g wood) followed by re-submerging in a shaking bath for 48 h [22].

The final delignified and relatively swollen wood fibers were washed to remove reagent residues, dissolved lignin, and hemicellulose. This was confirmed when a constant conductivity in the washing water was achieved, measured using a conductivity meter (S30 SevenEasy; Mettler Toledo, Schwerzenbach, Switzerland). The oxidized samples were then adjusted to identical concentrations of 0.2 wt%.

The suspensions (0.2 wt%) were then homogenized in one pass without recirculation using an APV-2000 high-pressure homogenizer (SPX Flow Inc., Silkesborg, Denmark) with an average flow rate of 4 mL/s at a pressure of 1000 bar. The suspensions were accurately standardized with respect to the solid content after homogenization prior to further characterization, and the conductivity of the CNF suspension was measured again after homogenization.

#### 2.2.2. Viscosity Measurement

The viscosities of the oxidized and homogenized suspensions were measured to evaluate any differences between the samples at various concentrations using a Vibro viscometer (SV-10, A&D Company Limited, Tokyo, Japan) with the tuning fork vibration method at a vibrational frequency of 30 Hz. The starting suspension was concentrated to 0.30 wt%, after which dilution was performed for each sample in 0.03 wt% increments down to 0.12 wt%.

#### 2.2.3. Carboxylate Content

The carboxylate content was analyzed using the electric conductivity titration method adapted from Saito and Isogai [9]. Further, 150 mL of CNF suspensions (≈0.2 wt%) were protonated with the addition of 0.1 M of hydrochloric acid and 0.01 M of sodium chloride. The suspensions were titrated with fresh 0.01 M of sodium hydroxide in 0.5 mL increments until a pH of 10 was reached. The number of carboxylate groups (mmol·g^−1^) induced by TEMPO oxidation was calculated using Equation (1).
(1)$$\mathrm{Carboxylate}\;\mathrm{groups}\;\left(\frac{\mathrm{mmol}}{g}\right)=\frac{C\left(V_{2}-V_{1}\right)}{m}$$
where *C* is the concentration of the sodium hydroxide, *V*_2_ and *V*_1_ are the volumes of added sodium hydroxide at the end and start, respectively, and *m* is the mass of the cellulosic material in the sample, calculated by subtracting the added acid, base, and salt mass from the oven-dried suspension. The measurements were repeated thrice, and the mean was taken.

#### 2.2.4. Atomic Force Microscopy

Atomic force microscopy (AFM) using a Veeco MultiMode scanning probe microscope (Santa Barbara, CA, USA) was conducted to confirm the presence of the nanofibrils and analyze their size. Antimony-doped silicon cantilevers (TESPA-V2, Bruker, Camarillo, CA, USA) with a spring constant of 42 Nm^−1^ and a nominal tip radius of 8 nm were used for the analysis. Samples were prepared by depositing a small droplet of the CNF suspension (0.001 wt%) on a freshly cleaved mica plate and letting it air dry for ≥5 h. The CNF width was measured from the height images to avoid broadening. Approximately 100 fully individualized CNFs from four AFM scans for each sample were analyzed using open-source software Gwyddion 2.61 and presented as the mean with the corresponding standard deviation.

#### 2.2.5. Nanofibril and Process Yield

The process yield was calculated as the washed gravimetric yield after the chemical treatment relative to the dry wood mass. The fraction of the process yield that comprised individual colloidally stable nanofibrils was further quantified by centrifugation of the suspensions at 12,000× *g* (Beckman Coulter J25i, Beckman Coulter AB, Bromma, Sweden) for 20 min at an approximate consistency of 0.2 wt%. The supernatant was decanted, and the solids retained in the sediment were dried for 24 h at 95 °C. This was repeated three times, and the nanofibril fraction was then calculated according to Equation (2).
(2)$$\Phi =1-\frac{m_{p}}{m_{p}+m_{s}}$$
where m_p_ and m_s_ are the dry precipitate and supernatant mass of the sample, respectively. The nanofibril fraction (Φ) was presented as a fraction of the process yield.

#### 2.2.6. Nanofibril Surface Area Estimation

The surface area of the nanofibrils was estimated in the never-dried state by analyzing the Congo red adsorption of the cellulosic fibers, as described by Spence et al. [26]. Briefly, the absorbed dye was quantified using a UV-vis spectrophotometer (GENESYS, 10 UV, Thermo Scientific, Schwerte, Germany) at an absorption maximum of 500 nm and then translated to the surface area (SSA) using Equations (3) and (4):(3)$$\frac{E}{A}=\frac{1}{K_{\mathrm{ad}}A_{\max}}+\frac{E}{A_{\max}}$$
(4)$$\mathrm{SSA}=\frac{A_{\max}\times N\times \mathrm{SA}}{\mathrm{MW}\;\times 10^{21}}$$
where E is the solution concentration of Congo red at equilibrium (mg/mL), A_max_ is the amount of Congo red absorbed in the sample (mg/g), K_ad_ is the equilibrium constant, N is Avogadro’s constant, SA is the theoretical surface area of a Congo red molecule (1.73 nm^2^), and MW is the molecular weight of Congo red (697 g/mol).

#### 2.2.7. Network Manufacturing and Characterization

The CNF suspensions were degassed for 30 min in a vacuum oven prior to vacuum filtration on hardened filter paper (Whatman, Grade 52, GE Healthcare, Machelen, Belgium, pore size: 7 µm). The wet networks were carefully peeled from the filter paper, dried to approximately 14% solid content, and characterized for water uptake or further pressed (2 kNm^−2^) for 10 h. The dry networks were finally compression-molded using Fontijne Grotnes LPC-300 (Vlaardingen, The Netherlands) between two mylar films (Lohmann Technologies, Knowl Hill, UK) at a pressure of 0.32 MPa and a temperature of 120 °C.

#### 2.2.8. Water Uptake

The wet networks were fully swelled in distilled water for 24 h prior to air drying at room temperature approx. 21–22 °C, where water uptake was studied as a function of time from an air-dried state to a fully hydrated state. Sections (0.1 g) of the hydrogels were collected and gravimetrically monitored over time.

#### 2.2.9. Mechanical Testing

The dry networks were cut into rectangular samples (40 × 5 mm) using a mechanical punch. The samples were then stored at 50% RH for at least two days prior to mechanical characterization. Mechanical testing was performed using a Shimadzu AG-X universal testing machine (Kyoto, Japan) with a 500 N load cell. Testing was performed at a crosshead speed of 10% min^−1^, and the strain was measured using a video extensometer (high-speed camera, HPV-X2). The gauge length was set to 20 mm for each measurement. Seven specimens were analyzed for each network batch. The tensile strength was reported as the maximum strength at break. The Young’s modulus was calculated from the slope of the linear (R^2^ = 0.95–0.99) portion in the elastic region (around 0.1–0.5% strain).

#### 2.2.10. Cellulose, Porosity, and Moisture Analysis

The cellulose content of the final networks was estimated by soaking in 17.5 M NaOH according to TAPPI 1999 [27], where the dissolved fraction was estimated as hemicellulose (and partly degraded low-molecular-weight cellulose). The porosity (P) of the networks was estimated using Equation (5):(5)$$P=1-\frac{\rho_{S}}{\rho_{C}}$$
where ρ_s_ is the density of the sample, and ρ_c_ is the theoretical density of cellulose (1.5 g·cm^−3^). The moisture content of the networks prior to mechanical testing was estimated as the difference in weight before and after 24 h of oven drying at 105 °C.

#### 2.2.11. Degree of Polymerization

The DP of the final CNFs (through networks) was estimated by calculating the intrinsic viscosity after the dissolution of the dry networks in a 0.5 M copper (II)-ethylenediamine complex and measurement using an ISO 17025 certified Ubbelohde viscometer. The viscosity at infinite dilution (limiting viscosity, [η]) was estimated according to TAPPI 1999 [28] and used to estimate the DP according to Equation (6):(6)$$\mathrm{DP}={\left(\frac{1.65\cdot \left[\eta \right]-116H\;}{C}\right)}^{1.11}$$
where H and C are the mass fractions of hemicellulose and cellulose, respectively. Furthermore, a DP of 140 was assumed for the hemicellulose fraction. The experiments were performed in triplicate for each sample.

## 3. Results

### 3.1. CNF Properties

The viscosities of the different CNF suspensions were evaluated at different concentrations of 0.30–0.12 wt% at 0.03 wt% intervals and are shown in Figure 1. As seen in Figure 1, the viscosities of T1 and T2 were lower than those of WT at higher concentrations. At 0.30 wt%, the viscosities plateaued at 52, 37, and 33 mPa·s for WT, T1, and T2, respectively. While the viscosities did not differ between the suspensions at lower concentrations, the presented viscosity trends indicate differences between the CNFs from T1 and T2 compared to WT. Factors that can influence viscosity include the aspect ratio of CNFs [29], the volume fraction of fine CNFs [30], and the ionic strength and pH [31,32].

To identify the cause of the lower viscosities in T1 and T2, the nanofibril yield, carboxylate content, suspension conductivity, and surface area were assessed, and the values are listed in Table 1. T1 and T2 exhibited lower suspension carboxylate content than WT. The process yields from the initial wood mass show no significant differences between the samples. The percentages of the yield corresponding to the fine nanofibrils (as determined by centrifugation) were similar for all samples at approximately 50 wt%. However, statistical analysis indicated a small difference between T1 and T2 compared to WT (Table 1). The suspension conductivity was indistinguishable for all the samples at approximately 40 µS/cm. The surface areas, as estimated using Congo red, which binds to the β-glucan surface, were lower for T1 and T2 compared to WT, with a difference in the surface area of approximately 20 m^2^·g^−1^.

These data suggest that the lower viscosities of T1 and T2 compared to WT observed in Figure 1 are attributable to a difference in the aspect ratio of the CNFs, which manifested as a slower increase in viscosity with an increasing concentration owing to a higher percolation threshold. The transition to a more viscous state at lower concentrations was demonstrated when comparing relatively short cellulose nanocrystals to CNFs, such as those isolated in this study [32].

The nanofibril yield indicates only a slight decrease for T1 and T2 compared to WT. Thus, it is likely that the fraction being discarded during the fine nanofibril yield calculation contributed to the increased fibrillation metrics because the small difference in the estimated CNF yield was unlikely to give rise to more distinct viscosity and surface area differences. Factors such as suspension concentration and centrifugation parameters affect the estimated fraction of nanofibrils. It was also shown that there is a level of experimental variation in the fine CNF yield for TEMPO-oxidized CNFs as a function of carboxylate content [33]. Based on the carboxylate content observed in this study, it is apparent that the chemical treatment resulted in the reduced oxidation of the retained solids for transgenics. This indicates that the CNFs of T1 and T2 were less susceptible to mild TEMPO oxidation, which is interesting because there were no differences in the chemical wood composition (further information is available in the Appendix A) [21].

The conductivity of the corresponding CNF suspensions was similar across all samples, indicating no differences in the ionic strength of the suspensions; therefore, appropriate filtration/purification was performed after TEMPO oxidation. The surface area was approximately 20% lower for T1 and T2 than for WT, which agrees with the lower carboxylate content observed in the transgenic samples. In accordance with previous reports, this is indicative of differences in cellulosic surface accessibility, which correlates with the degree of fibrillation [26,34]. Similar accessibility differences have been reported with increasing oxidation, where structural changes inside the nanofibril aggregate occur as chemical oxidation proceeds [35]. The CNF surface area estimated through the adsorption of dyes in this study is comparable to those reported in previous studies and ranges from 42 to 100 m^2^·g^−1^ and from 160 to 200 m^2^·g^−1^ [26,36,37].

To further elucidate the CNF characteristics, AFM analysis was performed, and the height scans of the nanofibrils are shown in Figure 2. The average CNF diameters measured from AFM height scans (Figure 2) did not reveal a statistical difference between the nanofibers, which were measured at 1.4 ± 0.3 nm, 1.4 ± 0.4 nm, and 1.9 ± 0.5 for WT, T1, and T2 CNFs, respectively. All CNFs were measured in the range of 0.5 to 3.5 nm which is indicative of surface peeling or splitting of the native elementary CNFs l (~3 nm) [38,39]. This can be explained by a disconnect between the degree of fibrillation and fibril height (at single-nanometer levels) [40], which makes the observed CNF diameter more complicated—significantly more than a generic fibrillation metric. An example of this effect is the observation of a lower CNF diameter for samples with less fibrillation overall [41].

Interestingly, shorter fibril fragments can be observed in the transgenic lines of the AFM scans in Figure 2 compared to the WT. This is exemplified in Figure 2 for the CNF from T1, where individual submicron fibrils predominate in contrast to WT, where the CNFs appear more seamless and interconnected. This could be a manifestation of the variation in the DP of cellulose that comprises these fibrils, where the DP is inherently correlated to nanofibril length [13]. This explains the difference in viscosity despite the relatively similar nanofibril yield, electron conductivity, carboxylate content, and volume fraction. This is further elaborated in the next section when analyzing corresponding CNF networks, where network behavior is significantly influenced by the DP of the CNFs that comprise them and is, thus, a good tool for assessing variation [12].

### 3.2. CNF Network Characterization

The manufactured networks were tested for their capacity to bind water prior to hot-pressing into dry networks. Water uptake is shown in Figure 3 together with the representative visual appearance of WT for all samples, before and after swelling, to the maximum capacity. Networks made from the CNF suspensions could absorb approximately 900–1100% of their initial weight, with a slightly lower capacity for T2 (~900 ± 30%) compared to WT (~1100 ± 40%). The capacity of T1 was indistinguishable from that of both WT and T2, where the values were within the error margins of T2 and WT. The difference between the WT and transgenic plants was supported by the slightly higher degree of carboxylation of the solids in the WT-CNF suspensions and their corresponding increased surface area (Table 1). The difference in water uptake and overall surface area can be associated with the lower carboxylate content in the transgenic CNFs than in WT-CNF. It is known that the CNFs with a higher degree of carboxylation show a higher affinity for swelling in water [42].

The tensile strength, strain, and Young’s modulus of the final dense and homogenous hot-pressed networks are shown in Figure 4a,b. The strength and maximum strain were lower for both transgenic samples T2 (125 ± 10 MPa; 5.7 ± 0.1%) and T1 (117 ± 10 MPa; 3.4 ± 0.7%) compared to WT (141 ± 14 MPa; 6.1 ± 0.2%). Interestingly, the stiffness, shown in Figure 4c, showed the opposite behavior with the lowest modulus for the WT (4.6 ± 0.3 GPa) compared to that of T1 (5.6 ± 0.5 GPa) and T2 (5.1 ± 0.1 GPa). However, when considering the standard deviations, no significant differences between the average values of T2 and WT were observed.

The decreased tensile strength and maximum strain for T1 relative to WT are supported by the data, where transgenic CNFs showed lower nanofibril yield, viscosity, degree of oxidation, and surface area (Table 1). All these factors can be connected to altered network performance [43]. The increased modulus for T1 compared to that of WT indicates an opposite trend and supports the interpretation of shorter CNFs in T1. The shortening of CNFs has been reported to increase the stiffness of the corresponding networks [12]. An extreme case of this effect was observed when comparing the stiffness of networks made from CNF suspensions obtained from harsh (TEMPO/NaBr/NaClO) and mild (TEMPO/NaClO/NaClO_2_) oxidation of wood powder, where the harsh treatment resulted in a network modulus that was approximately 30% higher compared to the one from the mild treatment [22]. Furthermore, the DP was already reduced in the initial transgenic wood cellulose, as determined by size-exclusion chromatography [21]. Thus, it is possible that the decreased native cellulose DP for the transgenic lines, as compared to WT, was preserved throughout processing and manifested in the final networks as decreased strain to break and increased stiffness for the corresponding networks.

To further elucidate the potential influence of DP on the mechanical behavior of the networks, the DP_V_ was estimated after dissolution in Cuen, a system compatible with TEMPO-oxidized CNFs [10,33]. The results are presented in Table 2, along with the moisture content, porosity, and α-cellulose content. The results indicate differences between WT and T1/T2, where CNFs from the transgenics had a lower DP compared to WT both before and after nanofibrillation. The moisture content, porosity, and cellulose content of the network were all relatively similar and, therefore, likely did not contribute to the difference in mechanical behavior.

The values obtained for DP_V_ agree with earlier work on wood cellulose after treatment with the TEMPO/NaClO/NaClO_2_ system [10]. The oxidative treatment in the present study had little effect on the initial DP, especially in comparison to the more rapid and aggressive TEMPO/NaBr/NaClO system [10]. In addition, the DP values appear to have a strong correlation to the length of the CNFs and therefore to the mechanical properties of the networks [13]. The difference in the DP of the feedstock is likely the strongest detectable distinction between the networks made from WT and T1/T2, and the findings on the mechanical properties in this study are supported by the literature [8,12,44], where brittle and stiff behavior is expected from a network made from CNFs composed of cellulose with a lower DP.

## 4. Conclusions

Wood from transgenic trees with reduced expression of *CSI1* was subjected to nanofibrillation using direct TEMPO oxidation (pH = 6.8), followed by high-pressure homogenization. The CNF suspensions and their corresponding networks were studied and compared to CNFs from wild-type wood (control). The CNF suspensions derived from transgenic trees show a lower degree of carboxylation, similar process yield, slightly decreased nanofibril yield, and a lower surface area. The viscosity decreased more rapidly as a function of concentration for the CNFs derived from transgenics, indicating a difference in the aspect ratio and capacity for network formation. This was further detected in manufactured networks, where the transgenic trees yielded CNF networks with reduced tensile strength, elongation at break, and DP, but with increased stiffness. Consequently, it is shown that the isolated CNFs and their network formation ability were influenced by the reduced expression of *CSI1* in the developing wood and augmented wood properties of the transgenics. Thus, this study shows that it is possible to influence the CNF quality by altering the wood properties of the feedstock and that the native cellulose structure influences the final isolated CNF properties.

## Figures and Tables

**Figure 1 nanomaterials-12-03448-f001:**
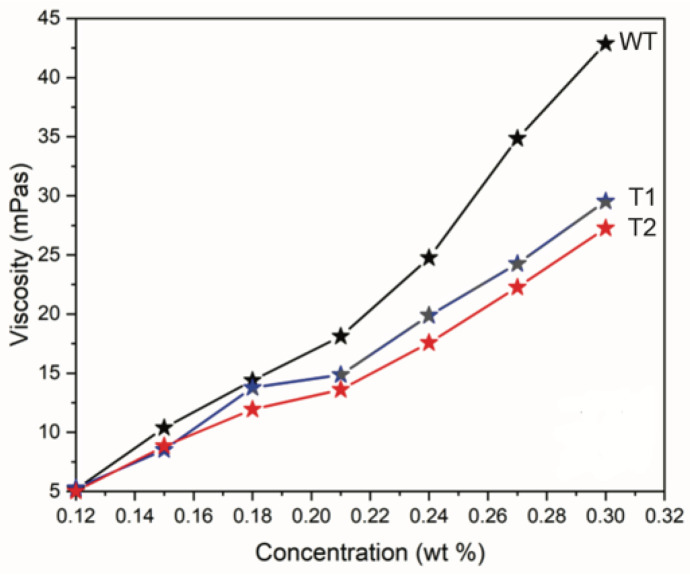
Viscosities as a function of suspension concentration for different CNFs from T1 and T2 compared to WT.

**Figure 2 nanomaterials-12-03448-f002:**
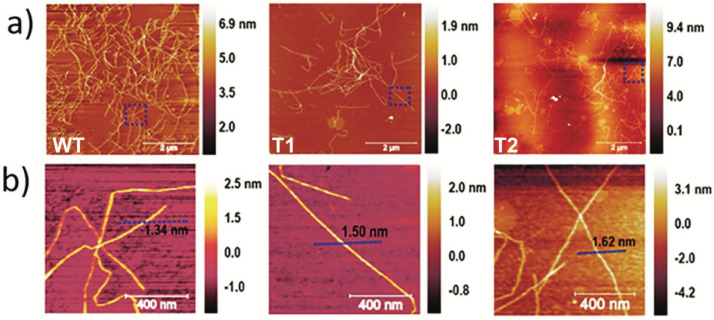
(**a**) Atomic force micrographs of CNFs from WT, T1, and T2. (**b**) Close-up of representative individual nanofibrils and measured height.

**Figure 3 nanomaterials-12-03448-f003:**
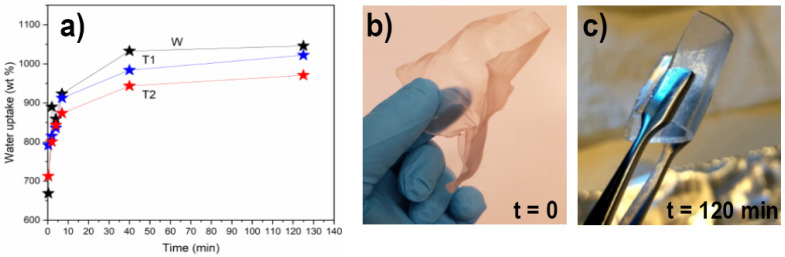
(**a**) Water uptake as a function of time for hydrogel networks made from the CNF suspensions. (**b**) Visual appearance of WT air-dried networks (>90% solid content); (**c**) Rehydrated networks of WT at maximum swelling (<8% solid content).

**Figure 4 nanomaterials-12-03448-f004:**
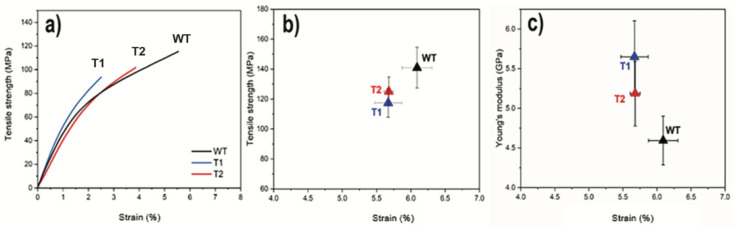
(**a**) Representative stress–strain curves of T1, T2, and WT; (**b**) Relation between tensile strength and strain; (**c**) Modulus and strain. (Error bars represent ± SD, n = 7).

**Table 1 nanomaterials-12-03448-t001:** Fibrillation metrics for CNFs isolated from WT, T1, and T2 with process yield, nanofibril yield, carboxylate content, viscosity (0.30 wt%), suspension conductivity (0.30 wt%), and surface area estimates of CNFs in suspension. Values in brackets indicate standard deviations.

Tree ID	Process Yield (wt%)	Nanofibril Yield (%)	Carboxylate Content (mmol/g)	Viscosity (mPa·s)	Suspension Conductivity (µS/cm)	Surface Area (m^2^·g^−1^)
WT	45.2 (1.3)a	50.8 (0.3)a	0.69 (0.10)a	51.8	37 (5)a	100
T1	44.0 (2.0)a	49.1 (1.0)b	0.55 (0.08)b	36.7	41 (8)a	83
T2	43.9 (2.1)a	48.2 (0.8)b	0.50 (0.06)b	33.3	42 (10)a	79

Mean values that do not share letters are significantly different according to Tukey’s test (*p* < 0.05).

**Table 2 nanomaterials-12-03448-t002:** Physical properties of the networks as determined prior to mechanical testing. Properties include average number degree of polymerization (DPn) for the initial wood cellulose (from Bünder et al., 2020), average viscosity degree of polymerization of final networks, moisture content, network porosity, and cellulose content.

Tree ID	Initial Cellulose DPn	CNF DPv	Moisture Content (%)	Porosity (%)	α-Cellulose (%)
WT	1802 (214)a	1185 (64)a	10.3 (2.1)a	22 (2)a	80
T1	1644 (137)b	995 (62)b	11.3 (2.8)a	21 (1)a	82
T2	1637 (129)b	1061 (37)b	9.7 (1.9)a	23 (2)a	80

Mean values that do not share letters are significantly different according to Tukey’s test (*p* < 0.05).

## Data Availability

Data will be made available upon reasonable request.

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
