# Peer review of "Characteristics of Cellulose Nanofibrils from Transgenic Trees with Reduced Expression of Cellulose Synthase Interacting 1"

_nanomaterials, 2022, doi:10.3390/nano12193448_

Round 1
Reviewer 1 Report
This paper is important for this field because it shows the possibility of controlling the properties of cellulose nanofibrils by genetic modification of the raw tree.
I recommend its publication in this journal with the following minor modifications.
1. The unit of suspension conductivity: please correct the discrepancy between Table 1 and the description in L231.
2. Is "Table 2" written in L300 an error for "Table 1"?
Author Response
Reviewer 1.
Response: Thank you so much for your careful reviewing of the paper we are grateful for your comments and corrections. We have revised the paper based on reviewers’ comments. The revision is highlighted with yellow.
Comment 1.The unit of suspension conductivity: please correct the discrepancy between Table 1 and the description in L231.
Response: Thank you so much for your careful reading and review , the unit in Table 1 is corrected to µS/cm.
Comment 2. Is "Table 2" written in L300 an error for "Table 1"?
Response: Thanks again for seeing this mistake, the Table 2 at L300 is corrected to Table 1.
Reviewer 2 Report
The authors presented cellulose nanofibrils from transgenic trees with reduced expression of cellulose synthase interacting 1 under direct TEMPO oxidation. The CNF suspensions and their corresponding networks were studied. It is interesting contribution. However, the manuscript should be revised before publishing on Nanomaterials considering the following remarks:
1.In figure 1, why was it selected for the concentration of the CNF suspensions between 0.12 and 0.30 wt%.
2. In figure and table, “WT”, “T1”, “T2”, should be marked.
3. From Figure 1, why did the difference of the viscosity of “WT”, “T1”, “T2” became greatly as the concentration of suspensions was above 0.21.
4. In line 300-301, the author presented that “The difference in water uptake and overall surface area can be associated with 300 the lower carboxylate content in the transgenic CNFs than in WT-CNF.” it should be carefully explained and cite references.
5. the tense of the manuscript is terrible in lines 234, 280, 329, 352, 354, and so on.
Author Response
Reviewer 2.The authors presented cellulose nanofibrils from transgenic trees with reduced expression of cellulose synthase interacting 1 under direct TEMPO oxidation. The CNF suspensions and their corresponding networks were studied. It is interesting contribution. However, the manuscript should be revised before publishing on Nanomaterials considering the following remarks:
We are thankful for the reviewer for the comments and help to improve the clarity of the paper. Below our responses and the revision highlighted with yellow color.
Comment: In figure 1, why was it selected for the concentration of the CNF suspensions between 0.12 and 0.30 wt%.
Response: Thanks for commenting this. The suspensions were fibrillated in the concentration of 0.2 wt%, already this concentration is resulting in a gel like material. These suspensions were concentrated to 0.3 wt% as starting concentration and the gel was diluted with 0.03 wt% increments to 0.12 wt% to study if the concentration affected the viscosity values. We can see that when the suspensions have low concentration the differences are small but when the concentration is increasing differences are seen, which depend on the nanofibers length etc. as explained in the text. The CNFs have small size and forms gel already in low concentration therefore these concentrations are max 0.3 wt%. We have now made that clearer in the text.
Comments: In figure and table, “WT”, “T1”, “T2”, should be marked.
Response: Thanks for pointing out the difficulty to see the difference of the materials. They were marked but it might be not possible to see the difference if the figures are not shown in color. (Black is WT, blue T1 and red T2). We have changed the figures and added the materials next to the graphs to improve the clarity. We have also revised the other figures which had same problem ( tensile strength, AFM).
Comment: From Figure 1, why did the difference of the viscosity of “WT”, “T1”, “T2” became greatly as the concentration of suspensions was above 0.21.
Response: We believe that this answer for the reviewer comment can be found in the text. ”The data suggest that the lower viscosity of T1 and T2 compared to WT observed in Figure 1 is attributable to a difference in the aspect ratio of the CNFs, which manifests as a slower increase in viscosity with increasing concentration owing to a higher percolation threshold. The transition into a more viscous state at lower concentrations has been demonstrated when comparing relatively short cellulose nanocrystals to CNFs, such as those isolated in this study [32]”
Comment: In line 300-301, the author presented that “The difference in water uptake and overall surface area can be associated with the lower carboxylate content in the transgenic CNFs than in WT-CNF.” it should be carefully explained and cite references.
Response: The answer for this comment is coming later in the text, we have tried to improve the clarification.
Comment: the tense of the manuscript is terrible in lines 234, 280, 329, 352, 354, and so on.
Response: We have tried to see the problems with the tense the reviewer is referring to but would like to explain that none of the authors are native English speaking and we need to relay on the language editing services. This text was corrected by a professional proof reading service, (Elsevier), a certificate is available. I suggest that if the language needs further to be improved, MDPI can do that if the paper is accepted to publication.
Reviewer 3 Report
This is a nice little study towards understanding further the influence of wood origin and species on the properties of extracted cellulose nanofibrils.
The manuscript is well written, and the rationale is clear and well defined. The study does bring some further insight on how genome-edited, modified trees can help decipher cellulose microfibrils arrangement in the cell wall and help produce CNFs with tailored performance.
I have not seen any specific typos, except Line 354 on page 9, where I believe the sentence is incomplete or missing something: "where the DP often approaches the level of DP".
I also have some specific questions on the scientific approach:
1- I question the use of TEMPO oxidation to compare wild type and transgenic feedstock. Why no using a mechanical treatment alone? TEMPO will modify the surface chemistry, acts on the hemicelluloses and lignin content in a way that may change the interactions of the CNF network.
2- As a follow up question, I suggest that the authors provide the chemical composition of their wild- and transgenic trees and resulting CNFs. What was the hemicelluloses and lignin content of your suspensions? How much did you lose in the process?
Although the authors specify that the trees had no significant differences in lignin and carbohydrates, this statement (line 90) does not imply that the composition of the CNFs will be the same. Adding this information is supporting information could be useful.
3- Another question relates to the discussion around the degree of polymerization. I feel that the authors based a lot their discussion and scientific discussion on the DP--- which is not necessarily an easy "reliable" characteristic to measure for CNFs.
A first question will be on the selection of the solvent and then comparison of DP and DPv. Do you expect the use of CED and Cuen to affect similarly the dissolution of your pulp and CNFs so that you can compare their DP?
Would the DP difference explain the differences in viscosity and mechanical properties? Is there another way to back up this conclusion?
Additionally, I wonder where you would find any difference in the crystalline phase of the fibrils? Have you measured at least the crystallinity index of your system? Did you see any differences?
4- The concept of apparent aspect ratio is mentioned, but the aspect ratio of the CNFs is never given. Did I miss this point?
Author Response
Reviewer 3. This is a nice little study towards understanding further the influence of wood origin and species on the properties of extracted cellulose nanofibrils. The manuscript is well written, and the rationale is clear and well defined. The study does bring some further insight on how genome-edited, modified trees can help decipher cellulose microfibrils arrangement in the cell wall and help produce CNFs with tailored performance.
Response: That you so much for your feedback and interesting comments. We are grateful for the time and efforts you have added to our work to improve its quality and clarity. The revision is highlighted with yellow.
Comment: I have not seen any specific typos, except Line 354 on page 9, where I believe the sentence is incomplete or missing something: "where the DP often approaches the level of DP".
Response: Thanks so much for seeing this mistake, we have revised the sentence.
Comment: I also have some specific questions on the scientific approach:
Comment: I question the use of TEMPO oxidation to compare wild type and transgenic feedstock. Why no using a mechanical treatment alone? TEMPO will modify the surface chemistry, acts on the hemicelluloses and lignin content in a way that may change the interactions of the CNF network.
Response: Thank you, we agree that the chemical treatment might affect the surface chemistry, but the mechanical fibrillation of wood is not possible, we have in our previous studies shown that the TEMPO oxidation we are doing is preserving the cellulose nanofibers better than other methods, Jonasson S, Bünder A, Niittylä T, Oksman K. Isolation and characterization of cellulose nanofibers from aspen wood using derivatizing and non-derivatizing pretreatments, Cellulose 27 (2019) 185. https://doi.org/10.1007/s10570-019-02754-w
We also think that as the chemical compositions of the initial wood samples are very similar and that the oxidation will affect the wood samples in a similar way.
Comment: As a follow up question, I suggest that the authors provide the chemical composition of their wild- and transgenic trees and resulting CNFs. What was the hemicelluloses and lignin content of your suspensions? How much did you lose in the process?
Response: This is an interesting question, and we agree that the exact chemical composition would increase knowledge on how the TEMPO oxidation will affect the composition. We did the cellulose content after the treatments, that is shown is Table 2, The cellulose content was rather similar for all the materials, around 80% and we expect that the remaining 20% is hemicellulose and the lignin is mainly oxidized. We can’t do the lignin content on these nanofibrils due to the lack of material but will do that in the future studies. The cell wall compositions of the used wood samples were characterized in the initial study where the effect of the CSI1 was studied on the wood, a Table of the wood cell wall compositions are presented is supporting information Table S1.
Comment: Although the authors specify that the trees had no significant differences in lignin and carbohydrates, this statement (line 90) does not imply that the composition of the CNFs will be the same. Adding this information is supporting information could be useful.
Response: We agree with the reviewer, unfortunately can’t make the lignin study but we have added the initial cell wall chemical compositions in the supporting information Table S1. If comparing the cellulose content which is approx. 40 % initially it increased to 80% after the oxidation and fibrillation, this is with the cost lignin and hemicelluloses, and we expect that the lignin is mainly oxidated, and lignin content will be minor compared to the hemicelluloses in the CNFs.
Round 2
Reviewer 2 Report
the tense of the manuscript should be checked carefully.
Author Response
Respected reviewer, thanks again for this comment, the manuscript was corrected by a professional proof reading service before the submission to Nanomaterials, I have attached the certificate.
However, if the language needs to be improved I suggest that it is done by the Editing service provided by MDPI.
